# Therapeutic Targeting of DNA Damage Response in Cancer

**DOI:** 10.3390/ijms23031701

**Published:** 2022-02-01

**Authors:** Wonyoung Choi, Eun Sook Lee

**Affiliations:** 1Research Institute, National Cancer Center, Goyang 10408, Korea; wonyoungchoi@ncc.re.kr; 2Center for Clinical Trials, National Cancer Center, Goyang 10408, Korea; 3Center for Breast Cancer, National Cancer Center, Goyang 10408, Korea

**Keywords:** DNA damage response, homologous recombination, BRCA, PARP inhibitor, ATM, ATR, DNA-PK, CHK1, CHK2, WEE1

## Abstract

DNA damage response (DDR) is critical to ensure genome stability, and defects in this signaling pathway are highly associated with carcinogenesis and tumor progression. Nevertheless, this also provides therapeutic opportunities, as cells with defective DDR signaling are directed to rely on compensatory survival pathways, and these vulnerabilities have been exploited for anticancer treatments. Following the impressive success of PARP inhibitors in the treatment of *BRCA*-mutated breast and ovarian cancers, extensive research has been conducted toward the development of pharmacologic inhibitors of the key components of the DDR signaling pathway. In this review, we discuss the key elements of the DDR pathway and how these molecular components may serve as anticancer treatment targets. We also summarize the recent promising developments in the field of DDR pathway inhibitors, focusing on novel agents beyond PARP inhibitors. Furthermore, we discuss biomarker studies to identify target patients expected to derive maximal clinical benefits as well as combination strategies with other classes of anticancer agents to synergize and optimize the clinical benefits.

## 1. Introduction

Human cancer is caused by genetic changes that cause uncontrolled cell proliferation [1]. Mutations occur as a result of DNA replication errors or inappropriately repaired DNA damage. Various cell-intrinsic mechanisms minimize the changes in DNA, and most steadily acquired somatic mutations are harmless; however, a small portion of these mutations can confer a selective advantage to the cell, which could allow the cell to grow or survive preferentially. The rates of mutational processes vary widely, but most human cancers carry 1000 to 20,000 point mutations and several hundreds of insertions, deletions, and rearrangements [2,3,4].

Accumulation of aberrant somatic mutations can trigger malignant cell transformation. Thus, human cells have evolved to prevent these errors via multiple layers of repair mechanisms. DNA damage response (DDR) pathways meticulously restore damaged sequences or direct irreparably damaged cells to undergo apoptosis or senescence. Genomic instability is an important hallmark of cancer, and defects in the DDR pathway may promote the growth of cancer cells by inducing de novo driver mutations, generating tumor heterogeneity, and evading apoptosis [1,5,6].

Ironically, however, defects in DDR machinery also provide therapeutic opportunities. As the pharmacologic blockade of the DDR signaling cascade directs tumor cells to rely on compensatory survival pathways, these vulnerabilities have been exploited in anticancer therapies [6]. The use of PARP inhibitors in *BRCA*-mutated cancers represents a successful example of this strategy, and more recently, the development of potent and selective agents that target DDR signaling components is rapidly emerging as a promising therapeutic option. 

Here, we summarize the key elements of the DDR pathway and how these molecular components may serve as anticancer targets. We also discuss the recent development of promising new agents that target the DDR pathway, biomarker studies to patient cohorts that are expected to derive clinical benefits from their use, and ongoing preclinical and clinical investigations to optimize these strategies.

## 2. Mechanisms of DNA Damage Response

Cells have various DNA repair mechanisms that act upon different kinds of genotoxic stress. In this section, we describe how various types of DNA damage are meticulously repaired, depending on the site of the damaged sequence and the type of damage.

### 2.1. Single-Strand Break

#### 2.1.1. Base Excision Repair (BER)

BER corrects small single-strand breaks (SSBs) that do not distort the DNA helix [7]. The first step of BER is to remove the damaged bases from the double helix. DNA glycosylase is a group of enzymes that recognize specific types of altered bases in DNA and cleaves the bond between deoxyribose and the damaged base. This creates an AP site (which stands for apurinic or apyrimidinic) that is recognized by AP endonuclease, removing the phosphodiester backbone. Then, poly (ADP-ribose) polymerase 1 (PARP1) and X-ray repair cross-complementing protein 1 (XRCC1) engage in the gap, promoting the assembly of repair factors. DNA polymerase β (Pol β) or DNA polymerase λ (Pol λ) conducts gap filling, followed by ligation by DNA ligase I or III [7,8,9].

#### 2.1.2. Nucleotide Excision Repair (NER)

NER is the main DNA repair system for the removal of bulky single-strand lesions that distort the DNA helical structure, mainly caused by ultraviolet irradiation or chemical compounds [10,11]. NER involves two major pathways: global genome NER (GG-NER) and transcription-coupled NER (TC-NER). GG-NER occurs anywhere in the genome, whereas TC-NER is responsible for the efficient repair of DNA lesions in the transcribed strand of active genes. GG-NER starts with recognition of helix-distorting lesions by the xeroderma pigmentosum C (XPC) complex, which is an indirect sensor for structural changes, and requires verification of whether the lesion should be repaired. TC-NER is initiated when RNA polymerase II stalls upon recognition of single-stranded lesions, causing helix-distortion [12]. These two subtypes of NER share a common pathway for further processing of DNA repair. The DNA-dependent ATPase/helicase transcription factor IIH (TFIIH) complex verifies the presence of damaged lesions. Subsequently, the structure-specific endonucleases, xeroderma pigmentosum group F protein (XPF), excision repair cross-complementation group 1 (ERCC1) complex, and xeroderma pigmentosum group G protein (XPG) incise the damaged strand, leaving a single-strand gap of 22–30 nucleotides. Finally, the gap is filled with DNA polymerase and ligase activities [11,13,14].

### 2.2. Double-Strand Break (DSB)

Damage to both strands of the DNA is especially dangerous because there is no intact template strand to ensure accurate repair. Ionizing radiation, replication errors, or reactive oxygen species may cause these types of DSBs. When left unrepaired, it quickly leads to chromosome breakdown. However, two distinct mechanisms, discussed below, have evolved to maintain genome integrity.

#### 2.2.1. Non-Homologous End Joining (NHEJ)

NHEJ is the simplest way to repair DSBs, as the broken ends are joined by DNA ligation with the resultant loss of nucleotides at the joining site [15,16,17]. Although it causes inevitable changes in the DNA sequence, most DSBs that occur in human cells are repaired by NHEJ [16]. The first step of NHEJ starts with the recognition of the DSBs by the Ku70-Ku80 heterodimer (Ku), which acts as a loading protein for recruiting other NHEJ components and promoting the repair process. Subsequently, DNA-dependent protein kinase (DNA-PK) catalytic subunit (DNA-PKcs) directly interacts with Ku to form the DNA–PK complex [15,16,17]. Because most DSBs result in two incompatible DNA ends with mismatching overhangs, subsequent end resection by endonuclease Artemis and nucleotide addition with DNA polymerases μ (Pol μ) and λ, (Pol λ) are necessary. Finally, the DNA ligase IV complex, consisting of X-ray repair cross-complementing 4 (XRCC4), XRCC4-like factor (XLF), and PAXX, a paralog of XRCC4 and XLF, performs the pivotal ligation step [15,16,17].

Due to its major role in repairing DSBs, loss of function in core NHEJ components has been demonstrated to drive carcinogenesis in murine models. Knockout mice of Ku70, Ku80, or DNA-PKcs are known to have a high incidence of lymphomas [18,19,20]. Nevertheless, the loss of core NHEJ components is rarely identified in human cancers. Unlike other organisms, Ku is considered an essential gene in humans as the genetic depletion of human cells with Ku subunits results in cell death, considered to be caused by the loss of telomere [21,22].

#### 2.2.2. Homologous Recombination (HR)

HR is a much more accurate way to repair DSBs than NHEJ as it precisely restores the original DNA sequence. DSBs can result from radiation or chemical damage but mostly arise from stalled DNA replication forks, irrespective of external stimuli. In general, sister chromatids are used as templates for the accurate repair of the original sequence. The damaged DNA has to be in close proximity to the template DNA for HR-directed repair to occur, and, for this reason, it occurs mainly in the S and G2 phases of the cell cycle when the two daughter DNA strands are sufficiently close to each other to serve as templates for each other [23]. In HR, the DSB ends are recognized by the meiotic recombination 11-RAD50-Nijmegen breakage syndrome 1 (MRN) complex, which initiates resection on one strand to generate 3′ single-stranded DNA (ssDNA) overhangs. Subsequently, the ssDNA-binding replication protein A (RPA) coats the overhangs of the breakage site, suppressing further resection. At this point, the DNA damage checkpoint machinery is switched on to arrest the cell cycle and stall the replication fork. The MRN complex at the DSB ends interacts with ataxia-telangiectasia mutated (ATM) kinase, and the RPA-coated ssDNA overhangs recruit and activate ATM and Rad3-related protein (ATR); both are key components in the regulation of the DDR [24]. The DNA strand exchange protein RAD51 subsequently binds to ssDNA to form a protein-DNA filament. This nucleoprotein filament binds to the template double-stranded DNA by stretching the duplex, pulling the strands apart and forming an intermediate structure known as the displacement loop (D-loop). At this point, breast cancer 1 (BRCA1), breast cancer 2 (BRCA2), and partner and localizer of BRCA2 (PALB2) proteins interact to promote RAD51 filament assembly and stimulate strand invasion [23]. The invading strand then searches for sequence homology by conventional base pairing, and, with an extended stretch of base pairing of at least 15 nucleotides, the invading strand is stabilized for DNA synthesis. After sufficient elongation, the invading strand is disengaged from the D-loop and returns to the ssDNA. The DSBs are then annealed with sequence homology created from the DNA synthesis steps within the D-loops [23,25].

Defects in the HR pathway have been highlighted, mostly with loss-of-function mutations of *BRCA1* and *BRCA2*, as germline mutations of these genes are significant risk factors for hereditary breast and ovarian cancers. BRCA1 and BRCA2 proteins are ubiquitously expressed without significant difference in the expression levels between human tissue types [26]. However, it has been highly intriguing to determine how these mutations predominantly transform cells in the breast and ovary. Although the exact molecular basis remains to be elusive, multiple lines of reports have suggested the role of hormones and tissue-specific DNA-damaging metabolites as the background for tissue-specific tumorigenesis [26,27].

### 2.3. DNA Damage Response Signaling 

To maintain genome stability, highly coordinated signaling pathways are activated upon DNA damage. DDR signaling recognizes DNA breaks and arrests cell cycle progression to promote DNA repair; alternatively, it activates pathways that lead to apoptosis or senescence in cases of extensive or irreparable damage. Upstream regulation of DDR signaling is mediated by ATM, ATR, and DNA-PKcs, which are phosphoinositide 3 kinase (PI3K)-related protein kinases (Figure 1).

ATM, a key regulator of DDR, is activated by DSBs and HR. The MRN complex at the DSB recruits and directly binds to and activates ATM, prompting a cascade of DDR signaling via its kinase activity. The most important step is phosphorylation of the Ser-139 residue of the H2A histone family member X (H2AX), forming γH2AX, which is required for the accumulation of chromatin remodeling complexes and DNA repair proteins. Subsequently, γH2AX recruits mediator of DNA damage checkpoint protein 1 (MDC1) and eventually triggers the phosphorylation-ubiquitylation cascade for DDR signaling [24,28]. ATM also regulates cell cycle checkpoints by the phosphorylation of checkpoint kinase 2 (CHK2) at the G2/M transition and p53 at the G1/S transition [29]. These mechanisms enable cell cycle arrest to promote DNA damage repair or lead to senescence or apoptosis in cases of irreversible damage.

ATR is an important DNA replication stress response kinase that is activated by a wide range of genotoxic stresses. ATR is recruited to DNA breakage sites via direct interaction with ATR-interacting protein (ATRIP), which recognizes RPA-coated ssDNA that arise from resected DSBs or NER intermediates. After localization to the breakage site, additional factors, such as DNA topoisomerase II binding protein 1 (TOPBP1) or Ewing’s tumor-associated antigen 1 (ETAA1), are required to activate ATR. Activated ATR phosphorylates numerous downstream molecules with checkpoint kinase 1 (CHK1), which is a key target for the regulation of checkpoint signaling [24,29,30]. Upon activation, CHK1 further phosphorylates cell division cycle 25 homolog A (CDC25A), leading to ubiquitination and degradation. As CDC25A is a phosphatase that removes inhibitory modifications from cyclin-dependent kinases (CDKs), CHK1 activation results in cell cycle arrest. CHK1 also phosphorylates and stabilizes WEE1, a kinase that inactivates CDK1 and CDK2. Therefore, the net effect of CHK1 kinase activation is cell cycle arrest at the G2/M checkpoint [30]. The Aurora kinase A (AURKA) and Polo-like kinase 1 (PLK1) axis adds another layer of G2/M checkpoint regulation [31]. AURKA and its cofactor BORA activate PLK1 by phosphorylating its Thr-210 residue, which then activates CDC25 and suppresses WEE1 to override the G2/M checkpoint [32,33,34]. Thus, the AURKA-PLK1 axis is inhibited upon DNA damage, serving as another important layer of cell cycle regulation. The DNA-PKcs is recruited to DSBs by the Ku heterodimer to form the protein complex DNA-PK, modulating the critical steps in NHEJ, as described in Section 2.2.1. The kinase activity of the DNA-PKcs suppresses DSB-induced and spontaneous HR, thereby directing the repair process towards NHEJ [35]. However, the DNA-PKcs also have an overlapping spectrum of substrates with ATM, including the key downstream target H2AX [36]. Therefore, DNA-PKcs could contribute to DDR signaling in ATM-deficient or-inhibited conditions, playing critical roles of redundancy.

## 3. Therapeutic Exploitation of DNA Damage Response

### 3.1. PARP Inhibitors

The PARP family of proteins plays a key role in the DDR. Following SSBs, these proteins bind tightly to DNA breaks, recruit DNA repair effectors, and remodel the chromatin structure around the damaged DNA [37]. The antitumor activity of PARP inhibition is based on the concept of synthetic lethality. In cells with defects in HR, initially manifested in *BRCA1* or *BRCA2* gene mutations, pharmacological inhibition of the compensatory DNA repair machinery leads to genomic instability, mitotic damage, and cell death [38,39,40]. PARP inhibitors block the catalytic activity by interacting with the binding site of nicotinamide adenine dinucleotide (NAD^+^), a PARP cofactor, in the catalytic domains of PARP1 and PARP2. However, the antitumor mechanism of PARP inhibitors is not confined to impeding the catalytic activity of the enzyme; they also cause the ‘trapping’ of PARP in a complex with the DNA, which interferes with the catalytic cycle of PARP1 and damages the genomic integrity [41]. The various clinically developed PARP inhibitors have similar catalytic inhibitory effects against PARP, but they differ in their PARP-trapping abilities (talazoparib > niraparib > olaparib = rucaparib > veliparib), which also correlate with their cytotoxic potencies [37,41,42,43].

Apart from DSBs that are left unrepaired, with PARP inhibitors in *BRCA*-deficient tumor cells as the main mechanism for synthetic lethality, recent studies have suggested an alternative mechanism in which replication gaps in SSBs are critical for sensitivity to PARP inhibitors [44,45]. PARP1 and BRCA proteins are functionally crucial in recruiting repair proteins at SSBs, and recent work by Cong et al. has demonstrated that replication-associated ssDNA gaps are the key factors that mediate the cytotoxicity of PARP inhibitors. While *BRCA1-* and *FANCJ*-deficient tumor cells are both defective in HR, only *BRCA1*-deficient cells were sensitive to olaparib treatment, and this difference stemmed from distinct replication fork lengthening, which reflects replication-associated ssDNA gaps [44]. Additionally, replication gap suppression was shown to confer resistance to PARP inhibitors as ATR inhibition, which accelerates ssDNA gap induction, restored sensitivity to olaparib in a PARP inhibitor-resistant cell line [44]. This model has added an additional layer of scientific knowledge in predicting responses in non-*BRCA* mutant cancers and understanding the resistance mechanisms of PARP inhibitors.

PARP inhibitors are the most extensively studied class of DDR inhibitors, and several PARP inhibitors have been approved for the treatment of ovarian and breast cancers by various regulatory officials, including the United States Food and Drug Administration and the European Medicines Agency. Currently, its treatment indications are expanding through various clinical investigations targeting beyond *BRCA1/2* mutations and a broader range of malignancies [6].

### 3.2. DDR Pathway Inhibitors beyond PARP Inhibitors 

DNA damage mostly occurs as SSBs, but DSBs are more detrimental to cells. Therefore, most DDR-targeted therapies have focused on altering the functions of repair machinery associated with DSBs or on inhibiting checkpoint molecules that act downstream of these repair processes.

#### 3.2.1. ATR Inhibitors

Given its critical role in the DDR pathway, ATR inhibition is a promising target for anticancer therapy. In preclinical studies, genetic inactivation by induced expression of a dominant-negative ATR kinase-dead mutant led to enhanced sensitivity to various anticancer agents, which provided the rationale for developing pharmacologic inhibition strategies [46,47]. Furthermore, the progress in high-throughput screening methods for ATR activity has enabled the discovery of potent and selective ATR inhibitors. 

VE-821 was one of the earliest selective inhibitors discovered via high-throughput screening of ATR activity; it showed the potent and more selective inhibition of ATR compared with the related kinases, ATM and DNA-PK [48]. It was optimized and modified to produce VE-822 and VX-970 with enhanced affinity to ATR and improved solubility, suitable for in vivo studies [49,50]. VX-970, also known as M6620 or berzosertib, is a first-in-class drug and has been studied as monotherapy or in combination with cytotoxic chemotherapeutic agents (topotecan, carboplatin, cisplatin, gemcitabine) in phase 1 clinical trials, which demonstrated its safety and clinical benefits (Table 1) [51,52,53,54]. These results led to further studies that tested berzosertib in combination with cytotoxic chemotherapy. In a randomized phase 2 trial for the treatment of platinum-resistant ovarian cancer, berzosertib combined with gemcitabine was associated with significantly prolonged progression-free survival (PFS) compared with gemcitabine alone (22.9 weeks vs. 14.7 weeks) [55]. Despite these promising results, clinical trials with negative results have also been reported. In a phase 2 trial for urothelial carcinoma therapy, berzosertib was added to gemcitabine and cisplatin as first-line treatment and compared with chemotherapy alone; PFS was not significantly different between the berzosertib group vs. chemotherapy alone (8.0 months for both arms), but overall survival (OS) was shorter in the berzosertib group (14.4 months vs. 19.8 months). The combination arm also had higher rates of serious adverse events, which were mostly related to myelosuppression [56]. Although the results of ongoing studies are anticipated, these phase 2 trial results imply that combination strategies should be optimized for the selection of partner drugs and target patients (Table 1).

AZD6738 (ceralasertib) was also developed using strategies stemming from high-throughput screening data. The initially discovered compound was AZ20, which inhibits ATR, with low IC_50_ values in various cell lines [57]. Nevertheless, its low aqueous solubility and weak inhibition of CYP3A4 are major hurdles for this drug to progress to clinical trials. However, chemical modifications that improve solubility and eliminate CYP3A4 inhibitory activity have facilitated the successful development of AZD6738 (ceralasertib), which is suitable for clinical studies [58]. Based on preclinical studies showing the antitumor activity of ceralasertib in combination with DNA-damaging anticancer agents, phase 1 trials were conducted on ceralasertib in combination with cytotoxic chemotherapeutic agents (Table 1). In a trial combining ceralasertib with paclitaxel, 57 patients with advanced solid tumors displayed good treatment tolerance, with dose escalation starting from 40 mg daily up to 240 mg twice daily (480 mg per day) on a 14-day schedule for each cycle. Encouraging antitumor activity was noted among patients with melanoma in the early cohorts, while enhanced effects were observed in later cohorts in patients with melanoma refractory to anti-cell death protein 1 (PD-1) and anti-programmed death-ligand 1 (PD-L1) treatments, evidenced by an overall response rate of 33.3% (*n*= 33) [59]. In another phase 1 trial, ceralasertib was combined with carboplatin under various dosage schedules. The starting dose of ceralasertib was 20 mg twice daily on a 17-day schedule but was later amended to shorten the treatment period to 10 days, then to 7, and finally to 2 days, as the participants were deemed intolerant to the treatment [60]. The maximum tolerated dose (MTD) was 40 mg once daily on days 1–2 with carboplatin (AUC 5) every 3 weeks. Among the 36 patients enrolled in this trial, two showed partial responses [60]. These two phase 1 trials, combining ceralasertib with different cytotoxic agents, suggested that ATR inhibitors may have appropriate partner drugs that could be combined to improve tolerability and antitumor efficacy. Further studies are ongoing, and the results are anticipated for the development of better combinatorial strategies. 

BAY 1895344 is another oral selective and potent ATR inhibitor discovered and optimized for in vivo studies using the high-throughput screening of chemical compounds [61]. Preclinical studies of this drug as monotherapy in DDR deficiencies revealed promising antitumor activity as well as synergistic activity in combination with DNA damage-inducing chemotherapy or radiotherapy [62]. In a phase 1 trial, 22 patients with advanced solid tumors or non-Hodgkin lymphoma refractory to standard treatments were enrolled (Table 1). BAY 1895344 was started at 5 to 80 mg twice daily, and the MTD was determined as 40 mg twice daily on a 3-days-on/4-days-off schedule. Among the 20 patients evaluable for tumor response, 4 achieved partial responses [63]. Interestingly, all four patients with partial responses had deleterious ATM mutations identified via targeted DNA sequencing or loss of protein expression based on immunohistochemistry [63]. As BAY 1995344 has shown promising antitumor activity as a single-agent therapy, further studies in combination with a PARP inhibitor (niraparib) or an immune checkpoint inhibitor (pembrolizumab) are underway (NCT04267939; NCT04095273). 

M4344 is an oral ATR inhibitor that is currently undergoing clinical studies. Preclinical data have shown its anticancer activity as monotherapy and its synergism with cytotoxic agents in organoid and xenograft models [64]. Currently, M4344 is undergoing phase 1 trials as a single agent and combined with carboplatin in advanced solid tumors (NCT02278250) and is also under preparation for trials in combination with niraparib for recurrent ovarian cancer therapy (NCT04149145).

**Table 1 ijms-23-01701-t001:** Clinical trials of ATR inhibitors.

Trial Phase	Disease Setting	Treatments	Most Common Grade ≥ 3 Toxicity	Efficacy	Reference
Berzosertib/M6620/VX-970					
1	Solid tumors	Escalating doses of M6620 with topotecan	Anemia (19%), Leukopenia (19%), Neutropenia (19%)	PR 2/21 (10%) SD 7/21 (33%)	[51]
1	Solid tumors	Escalating doses of M6620, or combination with carboplatin	Monotherapy: None With carboplatin: Neutropenia (22%)	Monotherapy: PR 1/17 (6%) SD 5/17 (29%) With carboplatin: PR 1/21 (5%) SD 15/21 (71%)	[52]
1	Solid tumors	Escalating doses of berzosertib with cisplatin	Neutropenia (20%), Anemia (17%)	PR 4/26 (15%) SD 15/26 (58%)	[53]
1	Solid tumors	Escalating doses of berzosertib with gemcitabine +/− cisplatin	With gemcitabine Neutropenia (16%) ALT increased (16%) Fatigue (16%) With gemcitabine + cisplatin Neutropenia (63%) Thrombocytopenia (38%)	With gemcitabine PR 4/48 (8%) SD 29/48 (60%) With gemcitabine + cisplatin PR 1/7 (14%) SD 4/7 (57%)	[54]
2	Ovarian cancer (Platinum-resistant)	Randomization (1:1) Gemcitabine +/− berzosertib	Gemcitabine + berzosertib: Neutropenia (35%) Anemia (15%) Gemcitabine alone: Neutropenia (28%) Anemia (11%)	Gemcitabine + berzosertib PFS 22.9 weeks Gemcitabine alone PFS 14.7 weeks (HR 0.57, 90% CI 0.33–0.98)	[55]
2	Urothelial carcinoma	Randomization (1:1) Gemcitabine + Cisplatin +/− berzosertib	Gemcitabine + cisplatin + berzosertib: Anemia (57%) Gemcitabine + cisplatin: Anemia (25%)	Gemcitabine + cisplatin + berzosertib: PFS 8.0 months OS 14.4 months Gemcitabine + cisplatin: PFS 8.0 months OS 19.8 months HR for PFS: 1.22 (95% CI 0.72–2.08) HR for OS: 1.42 (95% CI 0.76–2.68)	[56]
Ceralasertib/AZD6738					
1	Solid tumors	Escalating doses of berzosertib with paclitaxel	Neutropenia (30%) Anemia (23%)	CR 1/57 (2%) PR 12/57 (21%) SD 18/57 (32%)	[59]
1	Solid tumors	Escalating doses of berzosertib with carboplatin	Anemia (39%) Thrombocytopenia (36%) Neutropenia (25%)	PR 2/34 (6%) SD 18/34 (53%)	[60]
2	Melanoma	Ceralasertib + Durvalumab	Anemia (33%) Thrombocytopenia (23%)	PR 9/30 (30%) SD 10/30 (33%) Median PFS 7.1 months (95% CI 3.6–10.6) Median OS 14.2 months (95% CI 9.3–19.1)	[65]
BAY-1895344					
1	Solid tumors	Escalating doses of BAY-1895344	Neutropenia (55%) Leukopenia (18%) Thrombocytopenia (18%)	PR 4/21 (19%) SD 8/21 (38%)	[63]
M4344					
1	Solid tumors	Escalating doses of M4344 or combination with carboplatin	Trial ongoing (Not reported)	Trial ongoing (Not reported)	NCT 02278250

PR, partial response; SD, stable disease; PFS, progression-free survival; OS, overall survival.

#### 3.2.2. ATM Inhibitors

ATM is a master regulator of DSB repair and has been extensively explored as a therapeutic target for anticancer therapy. Preclinical studies have shown that inhibiting ATM kinase activity sensitizes cells to ionizing radiation [66]. The potent KU-55933, which was the first selective ATM inhibitor, induced significant sensitization to radiation and DNA-damaging chemotherapeutic agents, including etoposide, doxorubicin, and camptothecin [67]. Due to its high lipophilicity and low bioavailability, it was later optimized to KU-60019, with improved aqueous solubility [68]. However, the low bioavailability of these drugs makes them unsuitable for clinical studies. 

AZD0156 is a recently developed ATM inhibitor with high oral bioavailability [69]. Although AZD0156 alone was not effective in inhibiting the growth of cancer cells in a xenograft model of colon cancer (SW620) or patient-derived xenograft model of *BRCA2*-mutant breast cancer, the combination of AZD0156 with irinotecan (colon cancer) or olaparib (breast cancer) was shown to result in synergistic inhibitory effects [69]. Based on these preliminary data, AZD0156 is undergoing a phase 1 trial in combination with olaparib or irinotecan (NCT02588105).

Stemming from a similar developmental series as AZD0156, AZD1390 also displays high selectivity and potency against ATM kinase activity, along with good pharmacokinetic and pharmacodynamic profiles [70]. Additionally, this drug has been shown to display excellent blood–brain barrier (BBB) permeability. In preclinical studies, AZD1390 was shown to radiosensitize orthotopic mouse models of glioblastoma and lung cancer with brain metastasis [70]; consequently, it is currently undergoing phase 1 clinical trials in combination with radiation therapies for brain tumors (NCT03423628) and lung cancer (NCT04550104). 

M3541 is another recently developed oral ATM inhibitor [71]. Preclinical studies have shown that M3541 sensitizes human xenograft cancer models to ionizing radiation [71,72]. Thus, a phase 1 trial was conducted to test the antitumor activity of M3541 in combination with palliative radiotherapy (NCT03225105), but the results have yet to be published.

#### 3.2.3. DNA-PK Inhibitors

DNA-PK plays a critical role in NHEJ-mediated DSB repair. Inhibition of this upstream regulator has been explored as an attractive target for radio- or chemosensitization as these modalities, which constitute the backbone of anticancer treatment, frequently induce DSBs [73]. Due to its structural similarity to DNA-PK, early developmental attempts largely focused on inhibitors targeting PI3K.

In line with this background, CC-115 was developed during the optimization of the triazole-containing mammalian target of rapamycin inhibitors, which were later shown to confer DNA-PK inhibitory activity as well [74,75]. The first report of CC-115 in human cancer therapy was in patients with relapsed/refractory chronic lymphocytic leukemia (CLL) harboring ATM deletions/mutations; clinical benefits were observed in 7 out of 8 patients (Table 2) [76]. In a more recent study, CC-115 was tested in the dose-finding and cohort expansion phases and was well-tolerated and showed preliminary efficacy [77]. Partial responses were observed in patients with CLL (31%), and long-term disease stabilization was observed in patients with head and neck cancer, castration-resistant prostate cancer, and glioblastoma [77]. Ongoing studies of CC-115 for prostate cancer (NCT02833883) and glioblastoma (NCT02977780) are expected to provide additional insights into the clinical efficacy of this novel agent.

M3814 (nedisertib, or peposertib) is a selective and potent inhibitor of DNA-PK that has been shown to sensitize tumors to ionizing radiation and synergize with topoisomerase inhibitors in preclinical studies [78,79,80]. In the phase 1 trial, 31 patients with treatment-refractory solid tumors were enrolled in dose-escalation cohorts and administered M3814 as monotherapy, which displayed good tolerability and modest efficacy (stable disease was achieved in 12 out of 31 patients) [81]. Based on preclinical evidence demonstrating the synergistic antitumor efficacy of M3814 combined with ionizing radiation, numerous ongoing studies have investigated this treatment combination in rectal cancer (NCT03770689), glioblastoma and gliosarcoma (NCT04555577), prostate cancer (NCT04071236), pancreatic cancer (NCT04172532), head and neck cancer (NCT04533750), and neuroendocrine tumors (NCT04750954). It has also been tested in combination with pegylated liposomal doxorubicin in ovarian cancer (NCT04092270) and with avelumab plus radiation (NCT04068194, NCT03724890) in various solid tumors. 

AZD7648 is also a selective DNA-PK inhibitor that was screened and optimized among various chemical compounds developed by AstraZeneca [82,83]. Preclinical experiments have shown that AZD7648 enhances sensitivity to ionizing radiation and doxorubicin and synergizes with the PARP inhibitor, olaparib [82]. Based on these data, AZD7648 is undergoing a phase 1/2 trial that includes a monotherapy arm for dose-finding and a combination arm with pegylated liposomal doxorubicin (NCT03907969).

**Table 2 ijms-23-01701-t002:** Clinical trials of DNA-PK inhibitors.

Trial Phase	Disease Setting	Treatments	Most Common Grade ≥ 3 Toxicity	Efficacy	Reference
CC-115					
1	Refractory/relapsed CLL/SLL	CC-115 monotherapy	Not reported	PR 4/8 (50%) 1 PR with IWCLL criteria 3 PR with lymphocytosis SD 3/8 (38%)	[76]
1	Cohort A (dose-escalation): solid tumors Cohort B (dose-expansion): selected refractory solid tumors	CC-115 monotherapy	Cohort A: Patients with at least one related Gr 3 AE (41%) Cohort B: Patients with at least one related Gr 3 AE (26%)	Cohort A: CR 1/39 (3%) PR 1/39 (3%) SD 18/39 (46%) Cohort B: -CLL/SLL PR 5/16 (31%) SD 4/16 (25%) -Overall PR 7/78 (9%) SD 29/78 (37%)	[77]
M3814/Nedisertib/Peposertib					
1	Solid tumors	Escalating doses of peposertib	Maculo-papular rash (13%) Nausea (7%)	SD 12/31 (39%)	[81]
AZD7648					
1/2	Solid tumors	AZD7648 alone or in combination with pegylated liposomal doxorubicin	Trial ongoing (Not reported)	Trial ongoing (Not reported)	NCT 03907969

PR, partial response; SD, stable disease.

#### 3.2.4. CHK1/2 Inhibitors

CHK1 and CHK2 are the downstream targets of ATR and ATM, which are required for cell cycle arrest and DNA damage repair. Inhibition of these kinases results in premature entry into mitosis and accumulation of DNA damage, eventually causing cell death. Numerous inhibitors have been developed for this strategy. However, drugs including UCN-01 [84,85], AZD7762 [86,87], LY2603618 [88,89], MK-8776 [90], and GDC-0575 [91], have shown low or modest antitumor efficacy, precluding further clinical development. 

Prexasertib (LY2606368) is a CHK1 inhibitor that induces mitotic catastrophe and shows promising antitumor effects in preclinical models [92]. In phase 1 trials, treatment with prexasertib caused a high frequency of grade 4 neutropenia, ranging from 50% to 73%, although the effects were transient and reversible (Table 3) [93,94]. Phase 2 studies have been conducted in high-grade serous ovarian cancer [95], triple-negative breast cancer (TNBC) [96], and small cell lung cancer (SCLC) [97]. Although the efficacy was low for SCLC and modest for TNBC, prexasertib showed promising efficacy in ovarian cancer. In the phase 2 trial, 28 patients with *BRCA*-wild, recurrent high-grade serous ovarian cancer (enriched population with TP53 mutation) were enrolled, and 8 out of 24 patients with evaluable lesions showed partial responses [95]. Grade 4 neutropenia was also very common (79%) but was mostly transient, with a median duration of 6 days, and resolved without growth-factor support [95]. Recent studies have also tested the combination of prexasertib with standard chemotherapy [98], olaparib [99], or anti-PD-L1 antibody [100]. In the olaparib–prexasertib combination study, 29 patients were enrolled, 18 of whom had *BRCA1*-mutant high-grade serous ovarian cancer resistant to prior PARP inhibitors. However, prexasertib and olaparib resulted in partial responses in four patients (22%) and disease stabilization in six additional patients. Further studies of prexasertib are underway, both as monotherapy in TNBC (NCT04032080) and as combination therapy with cytotoxic agents, in small round cell tumor/rhabdomyosarcoma (NCT04095221) and brain tumors (NCT04023669).

#### 3.2.5. WEE1 Inhibitors

WEE1 is a kinase that negatively regulates the cell cycle by phosphorylating and inhibiting the activities of its substrates, CDK1 and CDK2 [101]. WEE1 stabilizes replication forks during the S-phase and activates the G2/M checkpoint, providing sufficient time for the DDR machinery to restore the error. Therefore, WEE1 inhibition drives cells into mitosis without proper DNA damage repair and increases replication stress, resulting in mitotic catastrophe [102]. 

Adavosertib (initially named MK-1775, then AZD1775 during development) was the first selective WEE1 inhibitor reported to have antitumor effects in p53-deficient cells in combination with DNA-damaging agents [102,103]. Furthermore, it is currently the only drug of this category that has been extensively studied in clinical trials (Table 4). 

In a phase 1 trial of adavosertib administered as monotherapy to 25 patients with refractory solid tumors, partial responses were observed in 2 patients, both of whom harbored tumors with a *BRCA1*-mutation [104]. In another phase 1 trial, adavosertib was administered as monotherapy or in combination with gemcitabine, cisplatin, or carboplatin [105]. Of the 176 patients enrolled in this study, 17 patients achieved a partial response, while patients with the *TP53*-mutation (*n* = 19) showed a response rate of 21% compared with 12% of patients with wild-type *TP53* (*n* = 33). However, 42% of patients treated with chemotherapy combinations had grade 3 or higher hematologic toxicities, which was the most common type of adverse event [105]. Based on these promising studies, further phase 1 studies have investigated the combination of certain anticancer agents in specific types of cancers. In locally advanced or unresectable pancreatic cancer, adavosertib was combined with gemcitabine and radiation therapy at various doses [106], and in locally advanced head and neck cancer, it was combined with concurrent chemoradiation therapy with cisplatin [107]. Both studies demonstrated good drug tolerability with encouraging clinical outcomes for this strategy, suggesting that further studies should be conducted on adavosertib in combination with chemotherapy and radiotherapy.

Adavosertib has been tested in various phase 2 trials targeting ovarian cancer [108,109,110,111], breast cancer [112], colon cancer [113], SCLC [114], uterine serous carcinoma [115], and in a basket trial based on DNA repair pathway mutations [116]. Promising antitumor effects were observed in patients with ovarian cancer and uterine serous carcinoma, but the results of studies in patients with colon cancer, breast cancer, and SCLC were discouraging. 

In studies of ovarian cancer, adavosertib was investigated in both platinum-resistant/refractory and platinum-sensitive conditions. In a single-arm phase 2 study, adavosertib was combined with carboplatin in *TP53*-mutated ovarian cancer refractory/resistant to platinum-based first-line treatment within three months, representing a patient population expected to have poor clinical outcomes. The response rate was 43%, with encouraging antitumor activity evidenced by a median PFS of 5.3 months and an overall survival of 12.6 months [108]. Another phase 2 trial that tested adavosertib for treatment of platinum-resistant/refractory ovarian cancer was designed as a four-arm study, combining adavosertib with either gemcitabine, paclitaxel, carboplatin, or pegylated liposomal doxorubicin; promising response rates were observed, in particular for an intense treatment regimen with the adavosertib–carboplatin combination (response rate of 66.7%), but at the cost of significant toxicity [111]. Recently, a double-blinded, randomized phase 2 trial demonstrated that combining adavosertib with gemcitabine resulted in significantly longer PFS and OS compared to gemcitabine alone in platinum-refractory/resistant ovarian cancer, enriched with *TP53* mutations [110]. In a single-arm phase 2 trial for uterine serous carcinoma after failure of first-line platinum-based treatment, adavosertib was administered as monotherapy with a response rate of 29.4% and a 6-month PFS rate of 47.1%, which met the primary endpoint of the study [115]. Overall, these results imply that adavosertib holds great potential as monotherapy or in combination with DNA-damaging agents, and the outcomes of ongoing studies are anticipated to broaden our clinical and mechanistic insights into WEE1 inhibition in various types of cancers.

**Table 4 ijms-23-01701-t004:** Clinical trials of WEE1 inhibitors.

Trial Phase	Disease Setting	Treatments	Most Common Grade ≥ 3 Toxicity	Efficacy	Reference
Adavosertib/MK-1775/AZD1775					
1	Solid tumors	Escalating doses of AZD1775	Lymphopenia (20%) Neutropenia (16%) Anemia (16%)	PR 2/25 (8%)	[104]
1	Solid tumors	AZD1775 alone or in combination with standard chemotherapy	AZD1775 + Gemcitabine: Neutropenia (33%) AZD1775 + Cisplatin: Neutropenia (12%) AZD1775 + Carboplatiin: Thrombocytopenia (31%) Neutropenia (18%)	AZD1775 + Gemcitabine: PR 4/81 (5%) AZD1775 + Cisplatin: PR 9/58 (16%) AZD1775 + Carboplatiin: PR 4/62 (6%)	[105]
1	Locally advanced pancreatic cancer	Escalating doses of AZD1775 with gemcitabine and radiation	Neutropenia (12%) Fatigue (9%) Fever (9%) Anorexia/Nausea/ Vomiting (9%)	Median OS 21.7 months (90% CI 16.7–24.8) Median PFS 9.4 months (90% CI 8.0–9.9)	[106]
1	Locally advanced head and neck cancer	Escalating doses of AZD1775 in combination with radiation	Lymphopenia (92%)	CR 8/10 (80%) PR 2/10 (20%)	[107]
2	TP53-mutated refractory ovarian cancer	AZD1775 in combination with carboplatin	Thrombocytopenia (48%) Neutropenia (39%)	CR 1/21 (5%) PR 8/21 (38%) SD 7/21 (33%)	[108]
2	TP53-mutated, platinum-sensitive ovarian cancer	Randomization (1:1) Paclitaxel and Carboplatin +/− AZD1775	AZD1775 + Chemotherapy: Neutropenia (36%) Placebo + Chemotherapy: Neutropenia (33%)	AZD1775 + Chemotherapy: PFS 7.9 months Placebo + Chemotherapy: PFS 7.3 months HR for PFS: 0.63 (95% CI 0.38–1.06)	[109]
2	Platinum-refractory ovarian cancer	Randomization (2:1) Gemcitabine +/− Adavosertib	AZD1775 + Gemcitabine: Neutropenia (62%) Placebo + Gemcitabine Neutropenia (30%)	AZD1775 + Gemcitabine: PFS 4.6 months Placebo + Gemcitabine: PFS 3.0 months HR for PFS: 0.55 (95% CI 0.35–0.90)	[110]
2	Platinum-resistant ovarian cancer	Adavosertib in combination with - Gemcitabine - Paclitaxel - Carboplatin - Pegylated liposomal doxorubicin (PLD)	with Gemcitabine Neutropenia (78%) with Paclitaxel Neutropenia (53%) with Carboplatin Thrombocytopenia (63%) with PLD Neutropenia (17%)	with Gemcitabine PR 1/9 (11%) with Paclitaxel PR 10/38 (26%) with Carboplatin PR 13/35 (37%) with PLD PR 3/12 (25%)	[111]
2	TNBC	Adavosertib in combination with cisplatin	Diarrhea (21%) Neutropenia (18%)	CR 3/34 (9%) PR 6/34 (18%) SD 13/34 (38%)	[112]
2	Colorectal cancer with *TP53* and *RAS* mutations	Randomization (2:1) (maintenance) Adavosertib or active monitoring	Adavosertib Diarrhea (9%)	Adavosertib PFS 3.6 months OS 14.0 months Active monitoring PFS 1.9 months OS 12.8 months HR for PFS: 0.35 (95% CI 0.18–0.68) HR for OS: 0.92 (95% CI 0.44–1.94)	[113]
2	Small cell lung cancer	Adavosertib monotherapy (biomarker-selected patients)	Diarrhea 1/31 (3%)	PR 0/31 (0%) SD 9/31 (29%)	[114]
2	Uterine serous sarcoma	Adavosertib monotherapy	Neutropenia (32%)	CR 1/34 (3%) PR 9/34 (26%)	[115]
2	Solid tumor with mutations in DNA repair genes	Adavosertib in combination with carboplatin	Anemia (39%) Thrombocytopenia (39%) Neutropenia (32%)	PR 0/24 (0%)	[116]

PR, partial response; SD, stable disease; TNBC, triple-negative breast cancer.

#### 3.2.6. PLK1 Inhibitors

PLK1 is a kinase that has an important role in overriding the G2/M checkpoint after DNA repair to re-enter the cell cycle. Since its activation promotes the re-entry of the cell cycle, PLK1 has been implicated to be overexpressed in a variety of cancers and thus serves as a potential therapeutic target.

Volasertib (initially named BI 6727) is an ATP-competitive inhibitor of PLK1 that was developed by modifying the previously developed BI 2536 by Boeringer Ingelheim (Table 5) [117]. In a phase 1 trial, volasertib was administered as monotherapy to 65 patients with refractory solid tumors, with doses escalating from 12 to 450 mg at a 3-week-interval schedule. Reversible hematologic toxicity was the main adverse event, which was mostly manageable. In terms of efficacy, partial response was shown in 3 patients, and stable disease in 26 patients (*n* = 65) [118]. In a single-arm phase 2 trial against urothelial cancer, volasertib monotherapy resulted in modest efficacy, with a median PFS of 1.4 months [119]. 

Volasertib was investigated in clinical trials against acute myeloid leukemia (AML) in both phase 2 and 3 trials. In a randomized phase 2 study, volasertib was tested in combination with low-dose cytarabine for AML patients unsuitable for induction chemotherapy. The primary endpoint was met as the rate of complete remission was superior with the volasertib combination (31% vs. 13%), but this was at the expense of hematologic toxicities [120]. However, in a phase 3 trial, this combination turned out to be unsuccessful, as volasertib in combination with low-dose cytarabine did not bring significant prolongation of overall survival despite the higher rates of complete remission [121].

Volasertib was also studied in solid tumors. In a phase 2 trial for the second-line treatment of non-small cell lung cancer (NSCLC), patients were randomized to three arms to either volasertib, pemetrexed, or a combination of the two agents [122]. The median PFS was 1.4 months for volasertib monotherapy, 3.3 months for volasertib combined with pemetrexed, and 5.3 months for pemetrexed alone, showing that volasertib has a minor role in the treatment of NSCLC. In a phase 2 trial of platinum-resistant ovarian cancer, patients were randomly assigned 1:1 to either volasertib or the investigator’s choice of non-platinum single-agent chemotherapy [123]. The 24-week disease control rate, which was the primary endpoint, was 30.6% for volasertib compared to 43.1% for chemotherapy. Together, these studies have shown that volasertib has limited antitumor efficacy in solid tumors against patient populations that are unselected for specific biomarkers. 

Onvansertib (also known as PCM-075 or NMS-1286937) is an orally available selective ATP-competitive inhibitor of PLK1 (Table 5) [124]. In a phase 1 trial, onvansertib was administered to patients with advanced solid tumors in cohorts of escalating doses. Hematologic events, including neutropenia and thrombocytopenia, were the major dose-limiting toxicities, and the best responses were stable disease in 5 out of 16 evaluable patients [125]. In a phase 1b trial, onvansertib was combined with either low-dose cytarabine or decitabine in patients with AML. Complete remission was observed in 5 out of 21 patients who were administered with the decitabine combination, which prompted further investigation in a phase 2 trial [126]. Currently, this drug is undergoing further clinical investigation in metastatic colorectal cancer with KRAS mutation (NCT03829410), pancreatic cancer (NCT04752696), and CRPC (NCT03414034). 

**Table 5 ijms-23-01701-t005:** Clinical trials of PLK1 inhibitors.

Trial Phase	Disease Setting	Treatments	Most Common Grade ≥ 3 Toxicity	Efficacy	Reference
Volasertib/BI 6727					
1	Solid tumors	Escalating doses of Volasertib	Neutropenia (14%) Thrombocytopenia (14%)	PR 3/65 (5%) SD 26/65 (40%)	[118]
2	Urothelial cancer	Volasertib 300mg every 3 weeks	Neutropenia (28%) Thrombocytopenia (20%)	PR 7/50 (14%) SD 13/50 (26%) Median PFS 1.4 months (95% CI 1.3–2.6) Median OS 8.5 months (95% CI 3.9–12.1)	[119]
2	AML ineligible for induction chemotherapy	Randomization (1:1) LDAC +/− Volasertib	Volasertib + LDAC: Febrile neutropenia (55%) LDAC: Febrile Neutropenia (16%)	Volasertib + LDAC: CR + CRi 6/45 (13%) LDAC: CR + CRi 13/42 (31%)	[120]
3	AML ineligible for induction chemotherapy	Randomization (2:1) Low-dose cytarabine +/− Volasertib	Volasertib + LDAC: Febrile neutropenia (59%) Thrombocytopenia (39%) Placebo + LDAC: Thrombocytopenia (29%) Febrile Neutropenia (28%)	Volasertib + LDAC: CR + CRi 123/444 (28%) Median OS: 5.6 months (95% CI 4.5–6.8) Placebo + LDAC: CR + CRi 38/222 (17%) Median OS: 4.8 months (95% CI 3.8–6.4) HR for OS: 0.97 (95% CI 0.8–1.2)	[121]
2	NSCLC, second-line treatment	Randomization - Volasertib - Pemetrexed - Volasertib + Pemetrexed	Volasertib Neutropenia (14%) Pemetrexed Fatigue (9%) Volasertib + Pemetrexed Neutropenia (11%)	Volasertib Median PFS: 1.4 months Pemetrexed Median PFS: 5.3 months Volasertib + Pemetrexed Median PFS: 3.3 months	[122]
2	Platinum-resistant ovarian cancer	Randomization (1:1) Volasertib vs. Chemotherapy (non-platinum)	Volasertib Neutropenia (44%) Chemotherapy Neutropenia (6%)	Volasertib 24 week DCR: 30.6% Chemotherapy 24 week DCR: 43.1%	[123]
Onvansertib/PCM-075/NMS-1286937					
1	Solid tumors	Escalating doses of NMS-1286937	Neutropenia (16%) Thrombocytopenia (16%)	SD 5/16 (31%)	[125]
1	AML	Escalating doses of onvansertib with either LDAC or decitabnie	Anemia (35%) Thrombocytopenia (25%) Neutropenia (25%)	Onvansertib + LDAC CR + CRi 1/15 (7%) Onvansertib + Decitabine CR + CRi 5/21 (24%)	[126]

CR, complete remission; CRi, complete remission with incomplete blood count recovery; PR, partial response; SD, stable disease; LDAC, low-dose cytarabine; DCR, disease control rate; AML, acute myeloid leukemia; NSCLC, non-small cell lung cancer.

## 4. Biomarkers for DDR-Targeted Therapies: Beyond *BRCA1/2* Mutations

Based on synthetic lethality, PARP inhibitors have shown impressive clinical responses, especially in patients with *BRCA1* or *BRCA2* gene mutations. However, patients without direct mutations in these genes but with defects in other DNA damage repair pathways are said to exhibit ‘BRCAness’, which may also indicate sensitivity to DDR-targeted therapies [127].

### 4.1. Homologous Recombination Deficiency (HRD) Scores

Given that cells with defects in HR are expected to display genomic instability, assays that detect these features would allow the identification of patients with BRCAness. To identify genomic ‘scars’ related to HR deficiency, previous studies quantified large-scale genetic changes to analyze and predict the *BRCA* gene mutation status. Loss of heterozygosity (LOH) was found to be more frequent in tumors defective in *BRCA1* or *BRCA2*, and the number of these regions was highly associated with *BRCA* gene mutation status [128]. Large-scale transitions (LST), defined as chromosomal breaks between adjacent regions of at least 10 MB, were found to be an indicator of *BRCA1/2*-inactivation status [129]. Telomeric allelic imbalance (TAI), which is the number of subchromosomal regions with allelic imbalance extending to the telomeric end of the chromosome, was inversely correlated with *BRCA1/2* expression [130]. Later, it was suggested that combining these factors by calculating the mean score of the three indices robustly predicted *BRCA1/2* deficiency in breast cancer [131]. Consequently, commercial tests, validated in clinical trials, are available for the estimation of BRCAness; these include myChoice CDx (Myriad Genetics), which combines tumor *BRCA* gene mutation analysis and the genomic instability score (unweighted sum of LOH, LST, and TAI) [132], and FoundationOne CDx (Foundation Medicine), which evaluates the percentage of genomic regions with LOH determined via next-generation sequencing [133,134]. PARP inhibitors demonstrated clinical benefits in patients without *BRCA1/2* mutations but with high HRD scores, emphasizing its predictive role in identifying patients with BRCAness [135,136]. However, the clinical validity of these HRD score tests, especially the cutoff values upon which treatment decisions are based, are mainly assessed in terms of PARP inhibitor responses rather than in terms of its biological status or predictive role in other DDR-targeted therapies; therefore, further validation is required for its application in a broader range of drugs [135]. 

### 4.2. Sequencing-Based Mutational Signatures

Genomic mutational signatures reflect nucleotide alterations caused by specific patterns of DNA-damaging insults [3]. In a seminal study that comprehensively classified somatic point mutations and large-scale genomic alterations from 7042 cancers into 20 distinct mutational signatures, ‘signature 3′ was highly associated with *BRCA1/2*-inactivating mutations [3,137]. This assay requires whole-exome or whole-genome sequencing, which limits its widespread utility as a clinical biomarker. However, recent developments in computational tools have allowed the detection of this signature using targeted panel sequencing [138]. Using this new method, signature 3 was validated to predict therapeutic responses to combined PARP and PD-1 inhibitor therapy in ovarian cancer [139]. Nevertheless, further studies are required to determine whether the assessment of signature 3 could also predict therapeutic responses to other DDR-targeted agents.

## 5. Combination Strategies of DDR-Targeted Therapies

In early clinical studies of DDR-targeted agents, durable clinical benefits were not achieved with single-agent therapies. In addition, sensitivity to these agents is inherently dependent on the DDR signaling pathway of the tumor. Thus, their combination with DNA damage-inducing therapies, including radiation or cytotoxic agents, has been widely adopted to maximize clinical benefits, as mentioned in the previous sections. Furthermore, novel combination partners, including DDR inhibitors of different classes, targeted agents, and immune checkpoint inhibitors, have also been tested to maximize the benefits of DDR-targeted therapies.

### 5.1. Combination with DDR Inhibitors

Preclinical studies have shown that concurrent targeting of multiple critical components of the DDR pathway leads to synergism and overcomes resistance to single-agent DDR inhibitors. This strategy has been mostly aimed at overcoming acquired resistance to PARP inhibitors, for which the best-known resistance mechanisms include restoration of HR repair and stabilization of the replication fork [140]. In preclinical studies, combining inhibitors of the ATR/CHK1/WEE1 pathway with PARP inhibitors proved to be an effective strategy [141,142,143]. Consequently, various clinical trials are ongoing to investigate combinations of DDR-targeted agents (NCT02588105, NCT03462342, NCT04149145, NCT03057145, and NCT04197713). However, overlapping toxicities of these drugs, most notably bone marrow suppression, remains a major challenge in the optimization of dosage schedules for clinical application.

### 5.2. Combination with Targeted Agents

Previous reports have demonstrated that pro-oncogenic signaling pathways can regulate DDR and cell cycle checkpoints through various mechanisms. Here, we summarize the strategies of combining DDR inhibitors with drugs that target oncogenic signaling cascades to maximize the antitumor effects. Most of the combination strategies are based on the addition of targeted agents to PARP inhibitors, the most extensively studied class among these drugs.

#### 5.2.1. Antiangiogenic Agents

Angiogenesis is a hallmark of cancer and essential for tumor growth and metastasis. Interestingly, the crosstalk between angiogenesis and DDR signaling was noted by observing that PARP inhibition leads to defects in angiogenesis and, conversely, that hypoxic tumor cells acquire HR defects, which lead to increased sensitivity to PARP inhibition [144,145]. These backgrounds justified clinical trials of anti-angiogenic agents and PARP inhibitors. Cediranib, a tyrosine kinase inhibitor of vascular endothelial growth factor receptor 1–3, has been investigated as a combination partner with olaparib, mainly in ovarian cancer. In a phase 2 trial, cediranib plus olaparib was compared to olaparib alone in platinum-sensitive, relapsed ovarian cancer and resulted in a significant increase of PFS (16.5 months vs. 8.2 months), including in germline *BRCA*-wild type/unknown patients [146]. In a randomized phase 2 trial for the treatment of platinum-sensitive recurrent ovarian cancer, the combination of niraparib with bevacizumab, a monoclonal antibody against VEGF-A, was associated with longer PFS compared with niraparib alone (11.9 months vs. 5.5 months) [147]. Based on these promising results, a phase 3 trial of olaparib plus cediranib in ovarian cancer is underway to validate the clinical benefits shown in early-phase studies [148]. However, the olaparib–cediranib combination showed discouraging results in a single-arm phase 2 trial in patients with pancreatic cancer without germline *BRCA* mutations, as no clinical responses were observed [149]. These data imply that appropriate patient and disease selection are important when applying these strategies in clinical settings.

#### 5.2.2. PI3K Inhibitors

PI3K signaling pathway activation plays an essential role in DSB sensing, and thus, combining PI3K and DDR inhibitors has been suggested as a potentially effective therapeutic strategy [150]. In preclinical studies, the combination of NVP-BKM120 (a PI3K inhibitor) and olaparib led to synergistic antitumor effects in mouse models [151,152]. This was followed by a phase 1 trial in which BKM120 with olaparib displayed promising clinical benefits in patients with recurrent ovarian cancer or TNBC, but also significant dose-limiting toxicities [153]. Alpelisib, an α-specific PI3K inhibitor, was also tested in combination with olaparib in a phase 1b trial targeting germline *BRCA*-mutant recurrent ovarian or breast cancer. The combination was both tolerable and effective, especially in epithelial ovarian cancer (*N* = 28), with 10 patients (36%) achieving a partial response and 50% showing stable disease [154]. In two phase 1 trials, the combination of olaparib with capivasertib, an AKT inhibitor, demonstrated tolerability and antitumor efficacy in both germline *BRCA1/2*-mutant and wild-type disease [155,156]. Further studies are ongoing (NCT04729387, NCT03660826) to validate the combination of PI3K-AKT pathway inhibitors and PARP inhibitors in various clinical situations.

#### 5.2.3. Antiandrogen Therapies

Androgen receptor (AR) signaling is the most important therapeutic target in prostate cancer, and emerging evidence suggests that AR also regulates a network of DNA repair genes [157,158]. In mouse models of prostate cancer, AR inhibition led to activation of the PARP pathway, and dual inhibition of AR and PARP led to synthetic lethality [159,160]. The combination of AR inhibition by enzalutamide (AR antagonist) and AZD7762 (CHK1/2 inhibitor) showed a synergistic effect in xenograft models of prostate cancer [161]. Based on these preclinical data, abiraterone was combined with olaparib or placebo in a randomized phase 2 trial in patients with metastatic castration-resistant prostate cancer (mCRPC); abiraterone plus olaparib was associated with a significant prolongation of PFS vs. abiraterone plus placebo (13.8 months vs. 8.2 months, respectively) [162]. Currently, a phase 3 trial to test this combination as first-line therapy in patients with mCRPC is underway to validate these findings in a larger population (NCT03732820).

#### 5.2.4. MAPK Pathway Inhibitors

The mitogen-activated protein kinase (MAPK) pathway includes the Ras-Raf-MEK-ERK cascade, and its activity is altered in many types of solid malignancies. Interestingly, a study that attempted to uncover the resistance mechanisms of PARP inhibitors noted that PARP inhibitors led to upregulation of MAPK signaling, and conversely, trametinib (a MEK inhibitor) led to upregulation of DDR signaling [163]. In addition, the combination of talazoparib and trametinib showed synergistic antitumor effects in a subset of ovarian cancer cell lines [163]. Based on this preclinical evidence, the combination of selumetinib (MEK inhibitor) and olaparib is undergoing a phase 1 trial (NCT03162627).

### 5.3. Combination with Immune Checkpoint Inhibitors

Increased tumor mutational burden is a surrogate marker for response to immune checkpoint inhibitors (ICIs) [164,165]. Tumors with DDR defects have a higher amount of accumulated somatic mutations, which suggests that they may show an enhanced response to ICIs. This concept is supported by the higher response rates to anti-PD1/PD-L1 therapies in patients with advanced urothelial cancers marked by DDR deficiencies [166]. Thus, DDR inhibitors have been suggested as promising candidate partners of ICIs, and these drug combination strategies are undergoing clinical investigation in a wide array of disease statuses [167]. 

Multiple lines of evidence have demonstrated the background mechanisms of how DDR inhibitors may potentiate antitumor immunity induced by ICIs. A number of studies have validated that PARP inhibitor-mediated DNA damage enhances T-cell recruitment and infiltration by activating the stimulator of interferon genes (STING) signaling pathway [168,169,170]. Accumulated DNA damage may lead to cytosolic DNA leakage, which activates the STING pathway, an innate immune cascade that boosts type 1 interferon signaling [171]. In addition, DSBs induced by X-rays or PARP inhibitors have been suggested to upregulate PD-L1 expression, which is a widely adopted biomarker for predicting the response to anti-PD1/PD-L1 antibodies [172,173]. 

A series of clinical trials have investigated the safety and efficacy of combining PARP inhibitors and anti-PD1/PD-L1 antibodies in various types of malignancies. The MEDIOLA trial (NCT02734004) is a phase 1/2 study investigating olaparib and durvalumab (anti-PD-L1 antibody) in four different types of cancers: (1) germline *BRCA*-mutated metastatic breast cancer [174], (2) germline *BRCA*-mutated platinum-sensitive relapsed ovarian cancer [175], (3) relapsed gastric cancer [176], and (4) relapsed SCLC [177]; this combination was tolerable, with no unexpected adverse events or additive toxicities observed. In the germline *BRCA*-mutated breast cancer cohort, the 12-week disease control rate (DCR) was 80% (24 out of 30 patients), meeting the primary endpoint [174]. In the germline *BRCA*-mutated ovarian cancer cohort, the 28-week DCR was 65.6% and the objective response rate (ORR) was 71.9% (23 out of 32 patients), including 7 patients who achieved complete remission [175]. However, the efficacy was not very promising in gastric cancer or SCLC cohorts, as the primary endpoints were not met [176,177]. Olaparib and durvalumab were also tested in an independent phase 1/2 clinical trial targeting mCRPC [178]. The median PFS for all 17 patients was 16.1 months, and 9 patients had radiographic and/or prostate-specific antigen responses. 

The TOPACIO trial (NCT02657889) is a phase 1/2 clinical study investigating the combination of niraparib and pembrolizumab (anti-PD1 antibody) in recurrent platinum-resistant ovarian cancer and metastatic TNBC [179,180]. This combination was tolerable, with no new safety signals reported for either cohort. In the ovarian cancer cohort, the ORR was 18% (11 of 60 evaluable patients), and the DCR was 65% (39 out of 60 patients) [179]. Notably, 79% (49 out of 60 patients with ovarian cancer) did not harbor tumor *BRCA1* or *BRCA2* (t*BRCA*) mutations, and 53% had a negative HRD status based on the assay from Myriad Genetics, but the clinical benefits of this combination were apparent regardless of t*BRCA* mutation or HRD status. In the TNBC cohort, the ORR was 29% (13 out of 45 patients) and the DCR was 49% (22 out of 45 patients), which also demonstrated the clinical benefit in patients with wild-type t*BRCA* [180]. 

Recently, ceralasertib (an ATR inhibitor) was combined with durvalumab in a phase 2 trial in patients with melanoma who had failed prior anti-PD1 therapy [65]. The ORR among evaluable patients was 30% (9 out of 30 patients) and the DCR was 63.3% (19 out of 30 patients). The response to the treatment combination was independent of prior immune checkpoint inhibitor responses, and biomarker analyses revealed that tumors with immune-enriched microenvironments or alterations in the DDR pathway had better chances of deriving clinical benefit [65]. 

Other DDR inhibitors are also undergoing active clinical investigations in combination with immune checkpoint inhibitors. As these two classes of drugs may show synergistic effects by targeting different tumor vulnerabilities, future trial results are anticipated to provide a broader view of therapeutic options [181].

## 6. Conclusions

Defects in DNA repair are abundant in cancer cells, offering an opportunity to exploit these alterations for clinical benefit. Although the molecular mechanisms of DDR have been an active area of scientific research for several decades, there remains more to understand in order to exploit this pathway as a therapeutic target. Following the success of PARP inhibitors, especially in the treatment of *BRCA*-mutated breast and ovarian cancers, biomarker studies to enrich the potential responders are an active area of research. In addition, optimal combination strategies with DNA-damaging cytotoxic agents, radiation, targeted agents, or ICIs are expected to broaden the range of indications of DDR-targeting strategies. The basic, preclinical, and clinical data discussed here emphasize the rapid growth of scientific knowledge in this field. Furthermore, ongoing investigations are expected to provide a more comprehensive understanding of this pathway and enable the development of better therapeutic strategies in the future.

## Figures and Tables

**Figure 1 ijms-23-01701-f001:**
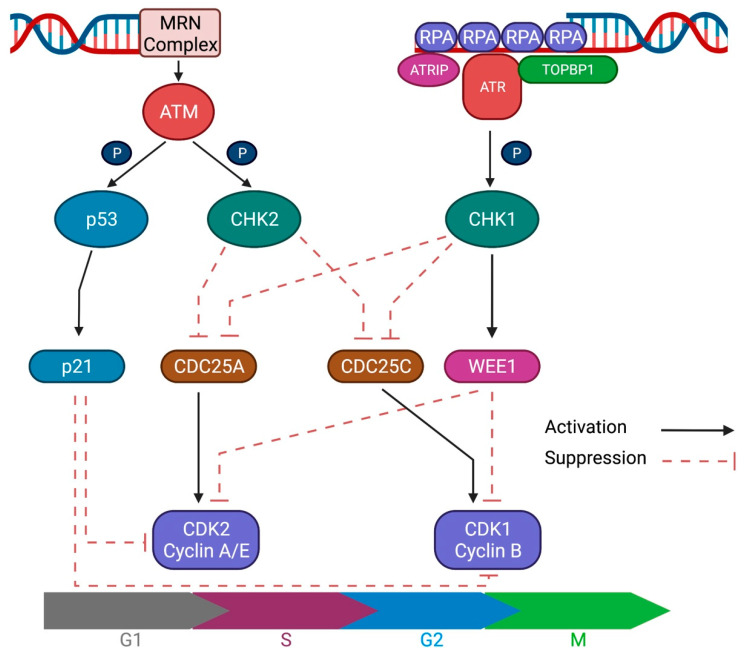
Schematic figure of ATM and ATR signaling. DNA damage response (DDR) signaling recognizes double-strand or single-strand breaks and arrests cell cycle progression to promote DNA repair. ATM and ATR are the key upstream regulators of the DDR signaling cascade.

**Table 3 ijms-23-01701-t003:** Clinical trials of CHK1/2 inhibitors.

Trial Phase	Disease Setting	Treatments	Most Common Grade ≥ 3 Toxicity	Efficacy	Reference
Prexasertib/LY2606368					
1	Solid tumors	Escalating doses of LY2606368	Neutropenia (89%) Leukopenia (71%) Anemia (69%)	PR 2/45 (4%) SD 15/45 (33%)	[93]
1	Solid tumors	Prexasertib monotherapy	Neutropenia (83%) Leukopenia (75%) Thrombocytopenia (33%)	SD 8/11 (73%)	[94]
1	Solid tumors	Prexasertib in combination with standard chemotherapy	Prexasertib + Cisplatin Neutropenia (67%) Prexasertib + Cetuximab Neutropenia (54%) Prexasertib + 5-FU Neutropenia (100%)	Prexasertib + Cisplatin PR 8/63 (13%) Prexasertib + Cetuximab PR 7/31 (5%) Prexasertib + 5-FU PR 1/8 (13%)	[98]
1	Solid tumors	Prexasertib in combination with olaparib	Neutropenia (79%)	*BRCA*-mutant HGSOC PR 4/18 (22%) SD 6/18 (33%)	[99]
1	Solid tumors	Prexasertib in combination with LY3300054 (anti-PDL1 antibody)	Neutropenia (82%) Leukopenia (76%)	PR 3/17 (18%) SD 8/17 (47%)	[100]
2	HGSOC (*BRCA*-wild type)	Prexasertib monotherapy	Neutropenia (93%) Leukopenia (82%)	PR 8/28 (29%) Median PFS 7.4 months (95% CI 2.1–9.4)	[95]
2	TNBC (*BRCA*-wild type)	Prexasertib monotherapy	Neutropenia (89%) Anemia (33%)	PR 1/9 (11%) SD 4/9 (44%)	[96]
2	Small-cell lung cancer	Prexasertib monotherapy	Neutropenia (65%)	Platinum-sensitive: PR 3/58 (5%) SD 15/58 (26%) Platinum-refractory: PR 0/60 (0%) SD 12/60 (20%)	[97]

PR, partial response; SD, stable disease; HGSOC, high-grade serous ovarian cancer; TNBC, triple-negative breast cancer.

## Data Availability

Not applicable.

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
