# Peer review of "Therapeutic Targeting of DNA Damage Response in Cancer"

_ijms, 2022, doi:10.3390/ijms23031701_

Round 1

Reviewer 1 Report

Choi and Lee revise our current knowledge and novel development of inhibitors of key DNA damage response factors, with a special interest in their therapeutic use in cancer. The authors provide a shallow introduction of the DNA repair mechanisms that maintain genome integrity, followed by a more comprehensive description of pharmacological inhibition of DNA repair kinases and proteins, as single or combined therapy, in the context of current ongoing clinical trials. The authors also compile the potential uses of DDR inhibitors in combination with other therapeutic strategies such as targeted, classical and immune therapies to improve the clinical benefits of these therapeutic approaches.

  • Point 1: The revision is quite comprehensive despite of lacking key information about the development of inhibitors targeting topoisomerases (key enzymes at relaxing the supercoiling induced by DNA or RNA machineries). I suggest the authors should emphasize the differences of “poisons” versus “catalytic” inhibitors.
  • Point 2: Mechanism of DNA damage response. The detailed description of each single DNA repair mechanism/pathway of genome maintenance is very simplistic and limited to very basic information. I would suggest the inclusion of information about differences and similarities, which might help to explain their tissue dependence. Why do MMR or BER factors (e.g. glycosylases, dealkylases…) are relevant to brain or colon tissues, whereas HR factors are to breast or ovarian cancers? Why does loss of NHEJ doesn´t impact on tumor progression despite of being a major DSB repair pathway whereas HR does?? I would also include schemes or drawings to facilitate their comprehension.
  • Point 3: The role of M phase checkpoint quinases and their inhibition are not described in the entire MS. The Aurora A or the PLK1 inhibitors (volasertib) for the treatment of haematological, lung or colon malignancies should be described, or at least cited.
  • I also miss the role of CDKs and their inhibitors as a single or in combination with other agents, in cancer progression. Flavopiridol as a first class inhibitors, and Dinacyclib could be discuss, and in combination with classical chemotherapy. Although some of these inhibitors main impact in RNA polII transcription and cell cycle regulation, some of them also regulate the DNA damage response.
  • Finally, a paragraph describing the control and maintenance of replication fork stability is also lacking. To this reviewer, this is crucial in relation to this topic, as the chemoresistance and sensitivity of tumor cells to PARP inhibitors seem to correlate with replication fork stability and persistance of single strand breaks at the forks. Please add a paragraph discussing this novel mechanism of fork instability (see Cong et al Mol Cell 2021; and related references).

Author Response

Please see the attachment for reviewer responses. 

Reviewer 2 Report

The manuscript summarizes the knowledge about the inhibitors of DNA repair pathways with potential or known use in cancer therapy. The manuscript is useful and well-written. There are several minor points that could further improve the presentation.

1) The list of DNA repair pathways in the introduction is incomplete. Kindly check and complement.

2) There are many references to old review papers. In addition, the readers would appreciate seeing the references to the original articles and, if not possible due to the limit of space, references to review articles published during the last 3-5 years. For example, in the introduction, each DNA repair sub-section is supplemented with several references, which sometimes appear to be random, neither original work, nor the newest papers; e.g. review articles from 2013.

Author Response

Please see the attachement for responses to the reviewer.

Round 2

Reviewer 1 Report

The MS has now included enough information about inhibition of key spindle checkpoint kinases.

I have no further comments